# Overweight or obesity in children born after assisted reproductive technologies in Denmark: A population-based cohort study

Kristina Laugesen[1]*, Katalin Veres[1], Sonia Hernandez-Diaz[2], Yu-Han Chiu[2,3], Anna Sara Oberg[2,4], John Hsu[5,6,7], Paolo Rinaudo[8], Mandy Spaan[9], Flora van Leeuwen[9], Henrik Toft Sørensen[1]

1 Department of Clinical Epidemiology, Aarhus University Hospital and Aarhus University, Aarhus, Denmark, 2 Department of Epidemiology, Harvard T.H. Chan School of Public Health, Boston, Massachusetts, United States of America, 3 Department of Public Health Sciences, Penn State College of Medicine, Hershey, Pennsylvania, United States of America, 4 Department of Medical Epidemiology and Biostatistics, Karolinska Institutet, Stockholm, Sweden, 5 Mongan Institute, Massachusetts General Hospital, Boston, Massachusetts, United States of America, 6 Department of Medicine, Harvard Medical School, Boston, Massachusetts, United States of America, 7 Department of Health Care Policy, Harvard Medical School, Boston, Massachusetts, United States of America, 8 Department of Obstetrics Gynecology and Reproductive Sciences, University of California, San Francisco, California, United States of America, 9 Department of Epidemiology, Netherlands Cancer Institute, Amsterdam, the Netherlands

* Kristina.laugesen@clin.au.dk

**Data Availability Statement:** Data cannot be shared publicly because of Danish legislation. Data

## Abstract

### Background

The association between assisted reproductive technologies (ARTs) and the body mass index (BMI) of children remains controversial. Confounding by morbidity and other factors associated with parental infertility may have biased studies comparing children born after ART with children born after no treatment. We investigated the associations between different fertility treatments and BMI in children at age 5 to 8 years, adjusting for and stratifying by causes of parental infertility.

### Methods and findings

This Danish cohort study included 327,301 children born between 2007 and 2012 (51% males, median age at follow-up 7 years). Of these, 13,675 were born after ART, 7,728 were born after ovulation induction with or without intrauterine insemination [OI/IUI], and 305,898 were born after no fertility treatments. Using the International Obesity Task Force (IOTF) standards, we defined overweight (BMI ≥ IOTF-25) and obesity (BMI ≥ IOTF-30). We compared children born after ART versus OI/IUI; intracytoplasmic sperm injection (ICSI) versus conventional in vitro fertilization (IVF); and frozen-thawed versus fresh embryo transfer and estimated crude and adjusted prevalences of children with overweight or obesity at age 5 to 8 years, prevalence odds ratios (PORs), and differences in mean BMI z-scores. Adjustment was performed using stabilized inverse probability of treatment weights, including parity, year of conception, parental causes of infertility, age, educational level, comorbidities, maternal country of origin, BMI, and smoking as covariates. The crude prevalence of obesity

are available through application to Danish health authorities (Statistics Denmark and the Danish Health Data Authority). The Danish Health Data Authority: www.sundhedsdatastyrelsen.dk, email kontakt@sundhedsdata.dk. Statistics Denmark: www.dst.dk, email dst@dst.dk.

**Funding:** This work was supported by a grant from NIH R01HD088393 received by JH. The funders had no role in study design, data collection and analysis, decision to publish, or preparation of the manuscript.

**Competing interests:** The authors have declared that no competing interests exist.

**Abbreviations:** ART, assisted reproductive technology; ATC, Anatomical Therapeutic Chemical; BMI, body mass index; CI, confidence interval; ICSI, intracytoplasmic sperm injection; IOTF, International Obesity Task Force; IPTW, inverse probability of treatment weighting; IUI, intrauterine insemination; IVF, in vitro fertilization; OI, ovulation induction; PCOS, polycystic ovarian syndrome; POR, prevalence odds ratio; SD, standardized difference.

was 1.9% in children born after ART, 2.0% in those born after OI/IUI, and 2.7% in those born after no fertility treatment. After adjustment, children born after ART and OI/IUI had the same prevalence of being overweight (11%; POR 1.00, 95% confidence interval [CI] 0.91 to 1.11; $p = 0.95$) or obese (1.9%; POR 1.01, 95% CI 0.79 to 1.29; $p = 0.94$). Comparison of ICSI with conventional IVF yielded similar pattern (POR 0.95, 95% CI 0.83 to 1.07; $p = 0.39$ for overweight and POR 1.16, 95% CI 0.84 to 1.61; $p = 0.36$ for obesity). Obesity was more prevalent after frozen-thawed (2.7%) than fresh embryo transfer (1.8%) (POR 1.54, 95% CI 1.09 to 2.17; $p = 0.01$). The associations between fertility treatments and BMI were only modestly different in subgroups defined by the cause of infertility. Study limitations include potential residual confounding, restriction to live births, and lack of detailed technical information about the IVF procedures.

## Conclusions

We found no association with BMI at age 5 to 8 years when comparing ART versus OI/IUI or when comparing ICSI versus conventional IVF. However, use of frozen-thawed embryo transfer was associated with a 1.5-fold increased risk of obesity compared to fresh embryo transfer. Despite an elevated relative risk, the absolute risk difference was low.

## Author summary

### Why was this study done?

- The number of children born after in vitro fertilization (IVF) is increasing.

- Information about long-term health of these children is limited, and the association between IVF and body mass index (BMI) of the children remains controversial.

- Various procedures, such as type of fertilization and embryo transfer, and underlying causes of infertility may influence associations.

### What did the researchers do and find?

- The researchers used Danish national registries to conduct a cohort study on the association between different fertility treatments and BMI in children at age 5 to 8 years, taking into account underlying causes of infertility.

- Children born after IVF had similar BMI compared with those born after ovulation induction (with or without intrauterine insemination) and compared with those born after no fertility treatment.

- Children born after frozen-thawed embryo transfer had higher BMI and higher prevalence of obesity compared with those born after fresh embryo transfer, compared with those born after ovulation induction (with or without intrauterine insemination), and compared with those born after no fertility treatment. Findings were consistent across subgroups defined by underlying cause of infertility.

**What do these findings mean?**

- Our study provides reassuring results for couples who seek help to conceive.

- Reasons behind our findings for frozen-thawed embryo transfer need further investigation.

- Longer follow-up and assessment of other outcomes related to metabolic health is needed.

- Study limitations include potential residual confounding, inclusion of live births only, and lack of detailed technical information about the IVF procedures.

## Introduction

Assisted reproductive technologies (ARTs), which include manipulation ex vivo of gametes and embryo like in vitro fertilization (IVF) or intracytoplasmic sperm injection (ICSI), have led to the birth of millions of children over the past decades [1,2]. Despite the increase in live births after these fertility treatments, information about long-term health consequences remains limited.

Examination of long-term outcomes has been challenging in part because of the need for substantial follow-up time, inclusion of large numbers of children born after treatment, and acquisition of granular clinical information about both parents and children. Further, it has been difficult to separate effects of treatment from underlying causes of infertility, i.e., indication for treatment. Although some studies have linked ART to adverse cardiometabolic health in children, including obesity [3–9], the association between ART and body mass index (BMI) of children remains controversial [3–6,10–21]. Divergent findings likely are explained by differences in study design, including different exposure definitions (varying ART procedures), reference groups (children born to parents with or without infertility), and approaches to controlling confounding. Recent studies have found an association between ART and lower BMI in childhood and early adolescence compared to being born after no fertility treatment, while the association attenuated or slightly reversed in late adolescence [20,21]. Associations persisted when children born after ART were compared to children of parents with infertility who did not receive treatment [20,21]. Comparison cohorts of children with parents with no need for reproductive assistance are helpful for contextualization, but confounding from underlying parental infertility remains an issue. Associations are best examined by comparing children born by parents with infertility requiring reproductive assistance. Because infertility itself is clinically heterogeneous, information about specific causes of infertility might be needed (but is often lacking), as it remains unknown if such causes confound or modify associations. Further, specific ART procedures might have different impacts on children. For instance, frozen-thawed embryo transfer is associated with a higher risk of being born large for gestational age, whereas fresh embryo transfer is associated with a higher risk of being born small for gestational age [6,15,22,23]. Also, studies indicate an association between frozen-thawed embryo transfer and higher BMI compared with fresh embryo transfer in childhood [3,6,15,20,21].

In this population-based cohort study based on national registry data, we aimed (1) to compare BMI among children aged 5 to 8 years born after ART versus ovulation induction with or without intrauterine insemination (OI/IUI); (2) to compare BMI after specific ART

procedures, such as ICSI versus conventional IVF or frozen-thawed versus fresh embryo transfer; and (3) to evaluate potential effect modification by underlying causes of infertility or biologic sex of the child.

## Materials and methods

This study is reported as per the Strengthening the Reporting of Observational Studies in Epidemiology (STROBE) guideline (S1 STROBE Checklist). The original analysis plan is shown in S1 Text. Some amendments were performed post hoc, including use of inverse probability of treatment weights for control of confounding and inclusion of additional potential confounders. Further, 5 extra sensitivity analyses were performed to assess robustness of our findings.

### Setting and data sources

This cohort study was conducted in Denmark (which has ≈65,000 births annually). The Danish healthcare system provides tax-supported health services to all residents, guaranteeing access to healthcare free of charge [24]. A unique personal identity number (the civil registration number) is assigned to all Danish residents at birth or upon immigration. The civil registration number is used in all registries, allowing accurate and unambiguous linkage. In Denmark, fertility treatment is paid by the public healthcare system (for the first child only) with some copayment for medical drugs. Currently, the system covers up to 3 rounds of OI/IUI plus 3 rounds of ART. Treatments are paid by couples when they wish to conceive a second or additional child (except if using frozen embryos from prior cycles), when more than 3 ART cycles are required to conceive, or when couples seek faster access than available from the public healthcare system. We used the following data sources.

- The Danish Medical Birth Register contains information on all births in Denmark since 1973. The information is collected by the midwife or physician overseeing the delivery and includes the civil registration number of the parents and infant, as well as characteristics of the pregnancy, the delivery, and the newborn [24]. The register does not include time to pregnancy.

- The Danish Fertility Treatment Registry contains information on all ART cycles since 1994 and on all OI/IUI cycles since 2006 conducted at public and private fertility clinics. ART covers conventional IVF and ICSI. The type of embryo preparation is also specified, such as fresh or frozen-thawed embryo transfer [25].

- The Danish National Children's Database records information on all children living in Denmark since 2011 and includes variables such as height, weight, and BMI categories. At age 5 to 8 years, children enroll in primary school and attend checkups performed by health nurses. All schools in Denmark report to the database, but approximately 0.1% of all Danish children do not attend primary school (they are homeschooled or do not attend any educational offer) [26]. Moreover, children attend a routine checkup at their general practitioner at age 5, and this is also reported to the database.

- The Danish National Patient Registry contains information on all hospital inpatient admissions since 1977. Outpatient clinic contacts and emergency department visits were added in 1995. For each inpatient, outpatient clinic, and emergency department discharge, 1 primary diagnosis and optional secondary diagnoses are recorded according to the International Classification of Diseases (ICD, 8th Revision between 1977 and 1993 [ICD-8] and 10th Revision [ICD-10] thereafter) [27].

- The Danish National Prescription Registry contains information on all prescriptions filled in community pharmacies since 1995. Filled prescriptions are coded according to the Anatomical Therapeutic Chemical (ATC) Classification System [28].

- The Danish social and demographic registries contain information on income and highest educational level of completed education.

- The Danish Civil Registration System contains information on date of birth, sex, and vital status for the entire Danish population since 1968. The system is updated daily [29].

## Study cohort

The Danish Medical Birth Register was used to identify members of the study cohort [24]. For this study, we identified 378,346 births in Denmark between January 1, 2007 and December 31, 2012. Stillbirths or those with duplicate civil registration numbers (1,580 [0.4%]) were excluded. A further 49,465 (13%) children were excluded who did not have routine anthropometric measurements at age 5 to 8 years recorded in the Danish National Children's Database. Thus, the final study cohort consisted of 327,301 children (Fig 1). Lack of routine anthropometric evaluation at age 5 to 8 years did not differ according to exposure or cause of infertility. Lack of BMI measurement was consistently 13% among children born after ART overall, conventional IVF, ICSI, frozen-thawed embryo transfer, fresh embryo transfer, or OI/IUI, and among children from the general population born after no fertility treatment. In addition, age at anthropometric evaluation was similar across groups (median age 7 years and interquartile range 6.6 to 7.6 years) (S2 Text).

## Assisted reproductive technologies and comparison cohorts

We established exposure and comparison cohorts through linkage between the Danish Medical Birth Register and the Danish Fertility Treatment Registry [25,30]. We assessed associations for ART versus OI/IUI, ICSI versus conventional IVF, and frozen-thawed versus fresh embryo transfer. Additionally, we compared frozen-thawed embryo transfer versus OI/IUI and fresh embryo transfer versus OI/IUI. For context and to compare with previous literature, we also assessed associations for children born after ART versus children from the general population born after no fertility treatment. This latter comparison reflects the combination of the treatment and the treatment indication (i.e., infertility) and not solely the treatment.

## Overweight, obesity, and body mass index

We assessed BMI at age 5 to 8 years using data recorded in the Danish National Children's Database. Our analysis was based on the International Obesity Task Force (IOTF) BMI definitions for overweight (BMI $\geq$ IOTF-25) and obesity (BMI $\geq$ IOTF-30) to categorize the children [31,32]. The IOTF age- and sex-specific cutoffs link BMI values at 18 years to child centiles based on data from 6 countries (Brazil, Great Britain, Hongkong, the Netherlands, Singapore, and the United States) fitted using the least mean squares method [32]. We further analysed BMI as a continous variable by using World Health Organization 2007 reference data and macro to calculate BMI z-scores [33].

## Covariates

For parents with children born after fertility treatment, we identified causes of infertility, including any female factor; ovulation disorders such as polycystic ovarian syndrome (PCOS); tubal factors; uterine or cervical factors such as endometriosis; nonspecific female factor; any

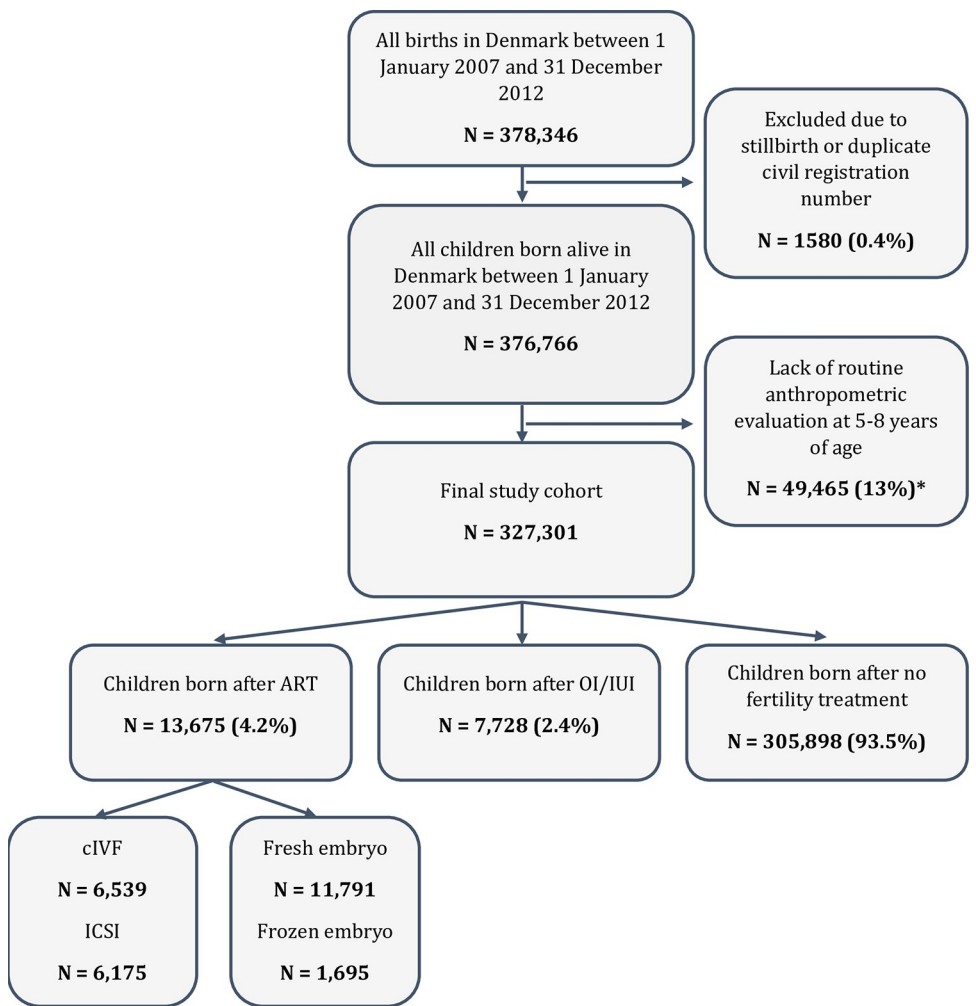

**Fig 1. Study cohort. Notes:** Abbreviations: ART, assisted reproductive technologies; cIVF, conventional in vitro fertilization; ICSI, intracytoplasmic sperm injection; IUI, intrauterine insemination; OI, ovulation induction. *Lack of routine anthopometric evaluation did not differ according to exposure or cause of infertility. Lack of anthropometric measurement was consistently 13% among children conceived with ART overall, conventional IVF, ICSI, frozen-thawed embryo transfer, fresh embryo transfer, or OI/IUI, and among children from the general population born after no fertility treatment. Further, children with and without anthropometric measurements had similar parental and birth characteristics (S2 Text).

male factor; and idiopathic causes of infertility. Most of this information was obtained from the Fertility Treatment Registry and supplemented with data from the Danish National Patient Registry (S3 Text). For the entire study population, we retrieved information from the Medical Birth Register on year of conception, parity, maternal and paternal age at conception, maternal BMI at pregnancy start, and maternal smoking status. As a measure of socioeconomic status, we obtained information from Danish social and demographic registries on the highest maternal and paternal educational level at conception (low [primary and lower secondary education], medium [upper secondary education or academy professional degree], or high [university education at the bachelor's degree level or higher]). From the Civil Registration System, we retrieved information on maternal country of origin (Nordic, European except for the Nordic countries, or non-European). Diabetes (types 1 or 2), hyperlipidaemia or use of lipid-lowering drugs, and hypertension or use of antihypertensive drugs were identified from

drug prescriptions or hospital records (inpatient and outpatient) in the Danish National Patient Registry or from National Prescription Registry records at any time before conception (S3 Text) [24]. Paternal BMI was recorded in the Fertility Treatment Registry starting in 2010. In context of potential mediation, we obtained information on gestational age at birth in weeks (<28, 28 to 31, 32 to 36, ≥37), birth weight, large for gestational age (defined as birth weight > the 90th percentile for infants of the same gestational age, sex, and birth year), small for gestational age (defined as birth weight < the 10th percentile for infants of the same gestational age, sex, and birth year), and singleton/multiple pregnancy.

## Statistical analysis

Among children born after fertility treatment, we described parental causes of infertility and treatment choices. We provided additional parental and birth characteristics among all children (i.e., born after fertility treatment or not). We further calculated the crude prevalence of overweight and obesity at age 5 to 8 years in children born after fertility treatment and in children from the general population born after no fertility treatment.

We compared children born after ART versus OI/IUI. Among children born after ART, we examined type of fertilization method, comparing children born after ICSI versus conventional IVF. We then focused on type of embryo transfer, comparing children born after frozen-thawed versus fresh embryo transfer. Additionally, we compared each type of embryo transfer to OI/IUI. We identified the following potential confounders from the literature: parental causes of infertility [34], maternal and paternal age at conception (restricted cubic splines with 3 knots) [35], maternal and paternal highest educational level at conception [36], maternal country of origin [37], maternal BMI (restricted cubic splines with 3 knots) [38], maternal smoking status [39], maternal and paternal hyperlipidemia or use of lipid-modifying drugs [40], maternal and paternal hypertension or use of antihypertensive drugs [41], presence/absence of diabetes (I or II) [42], parity [43], and year of conception [44]. To adjust for confounding, we used stabilized inverse probability of treatment weighting (IPTW) including the following steps [45]: For each child and each comparison, we used the listed potential confounders to compute a propensity score (i.e., the predicted probability of being exposed) using a logistic regression model and then graphically displayed the distribution of propensity scores across the various exposure and comparison cohorts (S1 Fig). We then used the propensity score to reweight the population by stabilized IPTW and truncated the extreme weights (the 1st and 99th percentiles) [45]. This approach created a population in which the treatment assignment was independent of the observed confounders, as these were balanced across the comparison groups. We assessed confounder balance before and after IPTW using standardized differences (SDs), with balance defined as a SD ≤0.15. Covariates occurring postconception (birth weight, etc.) might reside on the causal pathway and were therefore not adjusted for. Using ordinal logistic regression with robust variance (generalized estimating equations), we computed crude and adjusted (weighted) prevalence odds ratios (PORs) of overweight and obesity to estimate the average treatment effect in the population undergoing fertility treatment. To estimate crude and adjusted (weighted) differences in mean BMI z-scores, we used linear regression with robust variance (generalized estimating equations). Generalized estimating equations were used to account for family clustering (i.e., 1 woman could give birth to more than 1 child). We performed subgroup analyses by cause of infertility (if allowed by sample size) and sex of the child [46]. Propensity scores and weights were recalculated within each subgroup.

We conducted multiple sensitivity analyses to assess the robustness of our findings. First, to minimize confounding by treatment history, we considered only children born in the first

ART cycle. In a second sensitivity analysis, we included the previous number of OI/IUI and ART cycles in the IPTW. These 2 types of sensitivity analyses were applicable for comparisons of ART versus OI/IUI and ICSI versus conventional IVF. Because treatments involve both the mechanism of insemination (e.g., conventional IVF or ICSI) and the type of embryo transfer (e.g., fresh or frozen), we used restriction to enable single comparisons. For example, in a third analysis, we compared frozen-thawed versus fresh embryo transfer but restricted the cohort to children of parents undergoing conventional IVF only. Fourth, to enable comparison of ICSI versus conventional IVF not mediated through embryo transfer methods, we restricted the cohort to concern fresh embryo transfers only. Fifth, we separately assessed frozen-thawed embryo transfer in modified natural cycle (using hCG trigger only) and programmed cycle (estrogen and progesterone). Sixth, to evaluate the sensitivity of estimates to incomplete adjustment for confounding associated with parental factors, we added paternal BMI to the IPTW. This analysis was performed in a subpopulation of children born in 2011 and 2012 for ART versus OI/IUI in order to obtain a sufficient number of outcomes.

Last, in supplemental analyses, we compared children born after ART versus children from the general population born after no fertility treatment. For these comparisons, we used standardized morbidity ratio weighting instead of IPTW to estimate the average effects of treatment and treatment indication combined among the exposed.

Statistical analyses were conducted using SAS version 9.4.

## Ethics statement

This study was approved by the Danish Data Protection Agency (Record number: 2016-051-000001, serial number 1804). According to Danish legislation, informed consent or approval from an ethics committee is not required for registry-based studies.

## Results

### Parental causes of infertility and specific fertility treatments

Among the 327,301 children in our study cohort, 21,403 (6.5%) were born by parents with infertility using OI/IUI ($n$ = 7,728) or ART ($n$ = 13,675) (Fig 1). Causes of infertility varied among parents: ovulation disorders ($n$ = 4,221 children of which 58.9% were born after ART); tubal factors ($n$ = 3,222, 89.3% born after ART); cervical or uterine factors ($n$ = 2,557, 79.7% born after ART); male factors ($n$ = 13,097, 73.6% born after ART), and idiopathic causes of infertility ($n$ = 789, 26.1% born after ART). The type of fertility treatment differed by cause of infertility (Table 1).

### Parental and birth characteristics of children born after fertility treatment and no fertility treatment

Parental characteristics, such as age, educational level, maternal country of origin, comorbidities, and lifestyle, were similar for children born after ART versus OI/IUI, ICSI versus conventional IVF, and frozen-thawed versus fresh embryo transfer (Table 2). As expected, parental characteristics differed between children born after fertility treatment versus children born after no fertility treatment. Maternal and paternal age and educational level at conception were higher in children born after fertility treatment compared to children born after no fertility treatment (Table 2). Maternal smoking was less frequent among children born after ART (5.6%) and OI/IUI (5.7%) compared with children born after no fertility treatment (12.8%). Comorbidities, such as type 2 diabetes, hyperlipidemia/use of lipid-modifying drugs, and hypertension/use of antihypertensive drugs, were more frequent among parents in children

**Table 1. Parental causes of infertility\* and fertility treatment procedures among 21,403 children born to parents who received fertility treatment.**

| | N (%) | | | | | | |
|---|---|---|---|---|---|---|---|
| | Female factor (any) | Ovulation disorders | Tubal factor | Cervical or uterine factor | Nonspecific female factor | Male factor | Idiopathic |
| **All** | 17,538 (100) | 4,221 (100) | 3,222 (100) | 2,557 (100) | 3,808 (100) | 13,097 (100) | 789 (100) |
| **OI/IUI** | 6,224 (35.5) | 1,736 (40.1) | 346 (10.7) | 518 (20.3) | 1,108 (29.1) | 3,453 (26.4) | 583 (74.0) |
| **ART** | 11,314 (64.5) | 2,485 (58.9) | 2,876 (89.3) | 2,039 (79.7) | 2,700 (70.9) | 9,644 (73.6) | 206 (26.1) |
| **cIVF** | 6,073 (34.6) | 1,349 (32.0) | 1,958 (60.8) | 1,273 (49.8) | 1,604 (42.1) | 3,630 (27.7) | 136 (17.2) |
| **ICSI** | 4,379 (25.0) | 962 (22.8) | 682 (21.2) | 651 (25.5) | 907 (23.8) | 5,414 (41.3) | 70 (8.9) |
| **Fresh embryo transfer** | 9,704 (55.3) | 2,165 (51.3) | 2,425 (75.3) | 1,776 (69.5) | 2,318 (60.9) | 8,382 (64.0) | 187 (2.4) |
| **Frozen-thawed embryo transfer** | 1,426 (8.1) | 262 (6.2) | 418 (13.0) | 234 (9.2) | 321 (8.4) | 1,163 (8.9) | 17 (2.2) |

\*Causes of infertility are not mutually exclusive.

ART, assisted reproductive technologies; cIVF, conventional in vitro fertilization; ICSI, Intracytoplasmic sperm injection; IUI, intrauterine insemination; OI, ovulation induction.

born after ART and OI/IUI compared with children born after no fertility treatment (Table 2). Median maternal BMI was 23 kg/m$^2$ in all cohorts, although fewer mothers with children born after ART were obese (9.2%) compared with children born after OI/IUI (13.2%) or after no fertility treatment (11.6%) (Table 2).

The strongest predictors of overweight and obesity in children were maternal overweight or obesity, maternal country of origin, and low maternal and paternal educational levels (S4 Text).

Childbirth characteristics are shown in Table 2. The likelihood of being born small for gestational age was higher in children born after ART (13.9%) or OI/IUI (13.4%) compared with children born after no fertility treatment (9.2%) (Table 2). The prevalence was similar for ICSI (13.6%) versus conventional IVF (14.5%), but not for frozen-thawed (9.4%) versus fresh embryo transfer (14.4%). Conversely, being born large for gestational age was less prevalent in children born after ART (7.2%) or after OI/IUI (8.0%), compared with children from the general population (10.2%). With respect to specific types of ART, the likelihood of being large for gestational age was the same for ICSI (7.3%) versus conventional IVF (6.5%), but higher with frozen-thawed (11.4%) rather than fresh embryo transfer (6.6%).

## Crude prevalences of overweight and obesity among children age 5 to 8 years

Median age at follow-up was 7 years for all exposure and comparison cohorts. The crude prevalences of overweight were 10% in children born after ART, 11% in children born after OI/IUI, and 10% in children born after no fertility treatment (Fig 2). For obesity, the prevelances were 1.9%, 2.0%, and 2.7%, respectively (Fig 2). In children born after ART, the crude prevalence of obesity depended on type of fertilization (2.0% for ICSI and 1.6% for conventional IVF) (Fig 3) and type of embryo transfer (1.8% for fresh embryo transfer and 2.7% for frozen-thawed embryo transfer) (Fig 4).

**Table 2. Baseline characteristics of 327,301 children born in Denmark during 2007–2012 who underwent routine anthropometric evaluation at age 5–8 years.**

| | N (%) | | | | | | |
|---|---|---|---|---|---|---|---|
| | Children born after ART | | | | | Children born after OI/IUI | Children born after no fertility treatment |
| | **All ART** | **cIVF** | **ICSI** | **Fresh embryo** | **Frozen embryo** | | |
| Total | 13,675 (100) | 6,530 (100) | 6,175 (100) | 11,791 (100) | 1,695 (100) | 7,728 (100) | 305,898 (100) |
| Year of conception | | | | | | | |
| 2006–2009 | 8,589 (62.8) | 4,001 (61.3) | 3,620 (58.6) | 7,535 (63.9) | 945 (55.8) | 4,433 (57.4) | 209,881 (68.6) |
| 2010–2012 | 5,086 (37.2) | 2,529 (38.7) | 2,555 (41.4) | 4,256 (36.1) | 750 (44.2) | 3,295 (42.6) | 96,017 (31.4) |
| Previous number of OI/IUI cycles | | | | | | | |
| 0 | 8,748 (64.0) | 3,386 (51.9) | 4,652 (75.3) | 7,504 (63.6) | 1,099 (64.8) | 2,702 (35.0) | NA |
| 1 | 911 (6.7) | 476 (7.3) | 378 (6.1) | 781 (6.6) | 119 (7.0) | 1,896 (24.5) | NA |
| 2 | 808 (5.9) | 493 (7.5) | 267 (4.3) | 711 (6.0) | 91 (5.4) | 1,315 (17.0) | NA |
| ≥3 | 3,208 (23.5) | 2,175 (33.3) | 878 (14.2) | 2,795 (23.7) | 386 (22.8) | 1,815 (23.5) | NA |
| Previous number of ART cycles | | | | | | | |
| 0 | 4,689 (34.3) | 2,618 (40.1) | 1,802 (29.2) | 4,490 (38.1) | 127 (7.5) | 7,245 (93.8) | NA |
| 1 | 3,188 (23.3) | 1,433 (21.9) | 1,403 (22.7) | 2,499 (21.2) | 669 (39.5) | 162 (2.1) | NA |
| 2 | 2,290 (16.7) | 1,003 (15.4) | 1,131 (18.3) | 1,907 (16.2) | 368 (21.7) | 113 (1.5) | NA |
| ≥3 | 3,508 (25.7) | 1,476 (22.6) | 1,839 (29.8) | 2,895 (24.6) | 531 (31.3) | 208 (2.7) | NA |
| Oocyte donation | 187 (1.4) | 70 (1.1) | 102 (1.7) | NA | NA | NA | NA |
| Sperm donation | 547 (4.0) | 479 (7.3) | 62 (1.0) | 493 (4.2) | 53 (3.1) | 2,187 (28.3) | NA |
| Maternal characteristics | | | | | | | |
| Age at conception (median, IQR) | 33 (30–36) | 34 (31–37) | 33 (30–36) | 33 (30–36) | 33 (31–36) | 33 (30–36) | 30 (27–34) |
| Parity | | | | | | | |
| 0 | 8,573 (62.7) | 4,093 (62.7) | 3,873 (62.7) | 7,542 (64.0) | 905 (53.4) | 4,932 (63.8) | 129,856 (42.5) |
| ≥1 | 4,721 (34.5) | 2,259 (34.6) | 2,110 (34.2) | 3,903 (33.1) | 762 (45.0) | 2,616 (33.9) | 164,471 (53.8) |
| Missing | 381 (2.8) | 178 (2.7) | 192 (3.1) | 346 (2.9) | 28 (1.7) | 180 (2.3) | 11,571 (3.8) |
| Highest educational level | | | | | | | |
| Low | 1,010 (7.4) | 493 (7.5) | 444 (7.2) | 864 (7.3) | 124 (7.3) | 624 (8.1) | 51,656 (16.9) |
| Medium | 5,530 (40.4) | 2,567 (39.3) | 2,570 (41.6) | 4,776 (40.5) | 678 (40.0) | 2,952 (38.2) | 124,487 (40.7) |
| High | 6,894 (50.4) | 3,345 (51.2) | 3,054 (49.5) | 5,932 (50.3) | 875 (51.6) | 4,053 (52.4) | 118,737 (38.8) |
| Missing | 241 (1.8) | 125 (1.9) | 107 (1.7) | 219 (1.9) | 18 (1.1) | 99 (1.3) | 11,018 (3.6) |
| Country of origin | | | | | | | |
| Nordic | 12,474 (91.2) | 5,993 (91.8) | 5,605 (90.8) | 10,760 (91.3) | 1,545 (91.2) | 7,100 (91.9) | 259,342 (84.8) |
| European (except Nordic countries) | 457 (3.3) | 208 (3.2) | 211 (3.4) | 392 (3.3) | 56 (3.3) | 232 (3.0) | 15,278 (5.0) |
| Non-European | 637 (4.7) | 282 (4.3) | 307 (5.0) | 544 (4.6) | 83 (4.9) | 327 (4.2) | 26,743 (8.7) |
| Missing/unknown | 107 (0.8) | 47 (0.7) | 52 (0.8) | 95 (0.8) | 11 (0.6) | 69 (0.9) | 4,535 (1.5) |

*(Continued)*

**Table 2.** (Continued)

| | N (%) | | | | | | |
|---|---|---|---|---|---|---|---|
| | Children born after ART | | | | | Children born after OI/IUI | Children born after no fertility treatment |
| | All ART | cIVF | ICSI | Fresh embryo | Frozen embryo | | |
| BMI at start of pregnancy (kg/m$^2$) (median IQR) | 23 (21–26) | 23 (21–26) | 23 (21–27) | 23 (21–26) | 23 (21–26) | 23 (21–27) | 23 (21–26) |
| <18.5 | 448 (3.3) | 238 (3.6) | 181 (2.9) | 380 (3.2) | 64 (3.8) | 247 (3.2) | 12,050 (3.9) |
| 18.5–24 | 8,441 (61.7) | 4,202 (64.3) | 3,633 (58.8) | 7,260 (61.6) | 1,067 (62.9) | 4,441 (57.5) | 176,549 (57.7) |
| 25–29 | 2,854 (20.9) | 1,231 (18.9) | 1,409 (22.8) | 2,446 (20.7) | 365 (21.5) | 1,646 (21.3) | 59,509 (19.5) |
| 30–34 | 968 (7.1) | 397 (6.1) | 519 (8.4) | 847 (7.2) | 103 (6.1) | 702 (9.1) | 22,999 (7.5) |
| ≥35* | 291 (2.1) | 149 (2.3) | 126 (2.0) | 262 (2.2) | 26 (1.5) | 315 (4.1) | 12,463 (4.1) |
| Missing/ implausible | 673 (4.9) | 313 (4.8) | 307 (5.0) | 596 (5.1) | 70 (4.1) | 377 (4.9) | 22,328 (7.3) |
| Smoking during pregnancy | | | | | | | |
| Yes | 759 (5.6) | 397 (6.1) | 303 (4.9) | 660 (5.6) | 90 (5.3) | 441 (5.7) | 39,234 (12.8) |
| Missing | 399 (2.9) | 191 (2.9) | 188 (3.0) | 360 (3.1) | 28 (1.7) | 194 (2.5) | 12,985 (4.2) |
| Diabetes (type I, II) | 619 (4.5) | 315 (4.8) | 272 (4.4) | 536 (4.5) | 69 (4.1) | 574 (7.4) | 5,886 (1.9) |
| Type I diabetes | 71 (0.5) | 33 (0.5) | 37 (0.6) | 62 (0.5) | 8 (0.5) | 36 (0.5) | 1,297 (0.4) |
| Type 2 diabetes | 602 (4.0) | 282 (4.3) | 235 (3.8) | 474 (4.0) | 61 (3.6) | 538 (6.9) | 4,589 (1.5) |
| Hyperlipidaemia/ Lipid-modifying drugs | 99 (0.7) | 51 (0.8) | 39 (0.6) | 82 (0.7) | 17 (1.0) | 67 (0.9) | 1,542 (0.5) |
| Hypertension/ antihypertensive drugs | 1,681 (12.3) | 811 (12.4) | 765 (12.4) | 1,445 (12.3) | 197 (11.6) | 1,080 (14.0) | 32,480 (10.6) |
| Paternal characteristics | | | | | | | |
| Age at conception, (median, IQR) | 35 (32–39) | 35 (32–39) | 35 (32–39) | 35 (32–39) | 35 (32–39) | 34 (31–38) | 32 (29–36) |
| Highest educational level | | | | | | | |
| Low | 1,374 (10.0) | 605 (9.3) | 651 (10.5) | 1,161 (9.8) | 190 (11.2) | 745 (9.6) | 54,796 (17.9) |
| Medium | 7,067 (51.7) | 3,189 (48.8) | 3,350 (54.3) | 6,114 (51.9) | 860 (50.7) | 3,577 (46.3) | 155,469 (50.8) |
| High | 4,591 (33.6) | 2,300 (35.2) | 1,996 (32.3) | 3,947 (33.5) | 576 (34.0) | 2,553 (33.0) | 79,776 (26.1) |
| Missing | 643 (4.7) | 436 (6.7) | 178 (2.9) | 569 (4.8) | 69 (4.1) | 853 (11.0) | 15,857 (5.2) |
| BMI (kg/m$^2$)** | 25 (24–28) | 25 (23–28) | 25 (24–28) | 25 (24–28) | 25 (24–27) | 25 (24–27) | NA |
| Missing before 2010 | 8,916 (95.3) | 4.188 (96.0) | 3,792 (94.3) | 7,817 (95.6) | 983 (93.4) | 4,522 (91.9) | NA |
| Missing after 2010 | 2,908 (67.3) | 1,526 (70.5) | 1,380 (64.1) | 2,408 (66.7) | 445 (69.2) | 1,782 (63.5) | NA |
| Hyperlipidaemia/ Lipid-modifying drugs | 274 (2.0) | 112 (1.7) | 145 (2.3) | 226 (1.9) | 40 (2.4) | 129 (1.7) | 3,659 (1.2) |
| Hypertension/ antihypertensive drugs | 882 (6.4) | 411 (6.3) | 414 (6.7) | 756 (6.4) | 108 (6.4) | 424 (5.5) | 14,303 (4.7) |
| Birth characteristics | | | | | | | |
| Sex | | | | | | | |
| Male | 6,775 (49.5) | 3,337 (51.1) | 2,987 (48.4) | 5,868 (49.8) | 798 (47.1) | 3,923 (50.8) | 156,159 (51.0) |
| Gestational age, weeks | | | | | | | |

(Continued)

**Table 2.** (Continued)

| | N (%) | | | | | | |
|---|---|---|---|---|---|---|---|
| | Children born after ART | | | | | Children born after OI/IUI | Children born after no fertility treatment |
| | All ART | cIVF | ICSI | Fresh embryo | Frozen embryo | | |
| <28 | 89 (0.7) | 44 (0.7) | 37 (0.6) | 79 (0.7) | 10 (0.6) | 58 (0.8) | 566 (0.2) |
| 28–31 | 344 (2.5) | 189 (2.9) | 137 (2.2) | 301 (2.6) | 35 (2.1) | 137 (1.8) | 1,906 (0.6) |
| 32–36 | 1,283 (9.4) | 658 (10.1) | 535 (8.7) | 1,144 (9.7) | 123 (7.3) | 570 (7.4) | 8,975 (2.9) |
| ≥37 | 11,844 (86.6) | 5,581 (85.5) | 5,414 (87.7) | 10,162 (86.2) | 1,519 (89.6) | 6,896 (89.2) | 287,187 (93.9) |
| Missing | 115 (0.8) | 58 (0.9) | 52 (0.8) | 105 (0.9) | 8 (0.5) | 67 (0.9) | 7,264 (2.4) |
| SGA | 1,898 (13.9) | 946 (14.5) | 839 (13.6) | 1,702 (14.4) | 159 (9.4) | 1,032 (13.4) | 28,051 (9.2) |
| LGA | 979 (7.2) | 427 (6.5) | 449 (7.3) | 776 (6.6) | 193 (11.4) | 621 (8.0) | 31,190 (10.2) |
| Child from singleton pregnancy | 9,914 (72.5) | 4,768 (73.0) | 4,447 (72.0) | 8,515 (72.2) | 1,280 (75.5) | 6,239 (80.7) | 297,562 (97.3) |
| Child from multiple pregnancy | 3,761 (27.5) | 1,762 (37.0) | 1,728 (28.0) | 3,276 (27.8) | 415 (24.4) | 1,489 (19.3) | 8,336 (2.7) |

*Women with BMI above 35 are not treated with ART at public fertility clinics in Denmark.

**Available starting in 2010. Diabetes, hyperlipidaemia/lipid-modifying drugs, and hypertension/antihypertensive drugs are measured before conception.

ART, assisted reproductive technologies; BMI, body mass index; cIVF, conventional in vitro fertilization; ICSI, intracytoplasmic sperm Injection; IQR, interquartile range; IUI; intrauterine insemination; IVF, in vitro fertilization; LGA, large for gestational age; NA, not applicable; OI, ovulation induction; SGA, small for gestational age.

## The associations between fertility treatments and overweight, obesity, and BMI z-scores in children at age 5 to 8 years

After accounting for potential confounding, children born after ART or OI/IUI had the same prevalence of both overweight (11%; adjusted POR 1.00, 95% confidence interval [CI] 0.91 to 1.11; $p$ = 0.95) and obesity (1.9%; adjusted POR 1.01, 95% CI: 0.79 to 1.29; $p$ = 0.94), with minor fluctuations of point estimates within subgroups defined by underlying causes of infertility (Fig 2). Differences in mean BMI z-scores between children born after ART versus OI/IUI supported these findings, by yielding a null result (adjusted mean difference of −0.01, 95% CI: −0.05 to 0.02; $p$ = 0.45) (Fig 5).

Comparison of ICSI with conventional IVF yielded adjusted PORs of 0.95 (95% CI: 0.83 to 1.07; $p$ = 0.39) for overweight and 1.16 (95% CI: 0.84 to 1.61; $p$ = 0.36) for obesity (Fig 3), and a difference in the adjusted mean BMI z-score of 0.01 (95% CI: −0.03 to 0.06; $p$ = 0.51) (Fig 5). The association with obesity was modified by underlying causes of infertility, with adjusted PORs ranging from 0.89 (95% CI: 0.49 to 1.62; $p$ = 0.70) for ovulation disorders to 1.41 (95% CI: 0.66 to 2.98; $p$ = 0.37) for the nonspecific female factor category (Fig 3).

Frozen-thawed versus fresh embryo transfer was associated with obesity both before and after accounting for confounding with an adjusted POR of 1.52 (95% CI: 1.09 to 2.17; $p$ = 0.01) (Fig 4). This association remained consistent across all underlying causes of infertility (Fig 4). The mean BMI z-score was higher in children born after frozen-thawed versus fresh embryo transfer [adjusted mean difference = 0.07 (95% CI: 0.01 to 0.13; $p$ = 0.02)] (Fig 5). The frozen-thawed versus fresh embryo association was mainly driven by higher BMI and risk of obesity among children born after frozen-thawed embryo transfer. This was shown by comparing fresh embryo transfer with OI/IUI (adjusted POR of obesity of 0.96, 95% CI: 0.75 to 1.24; $p$ =

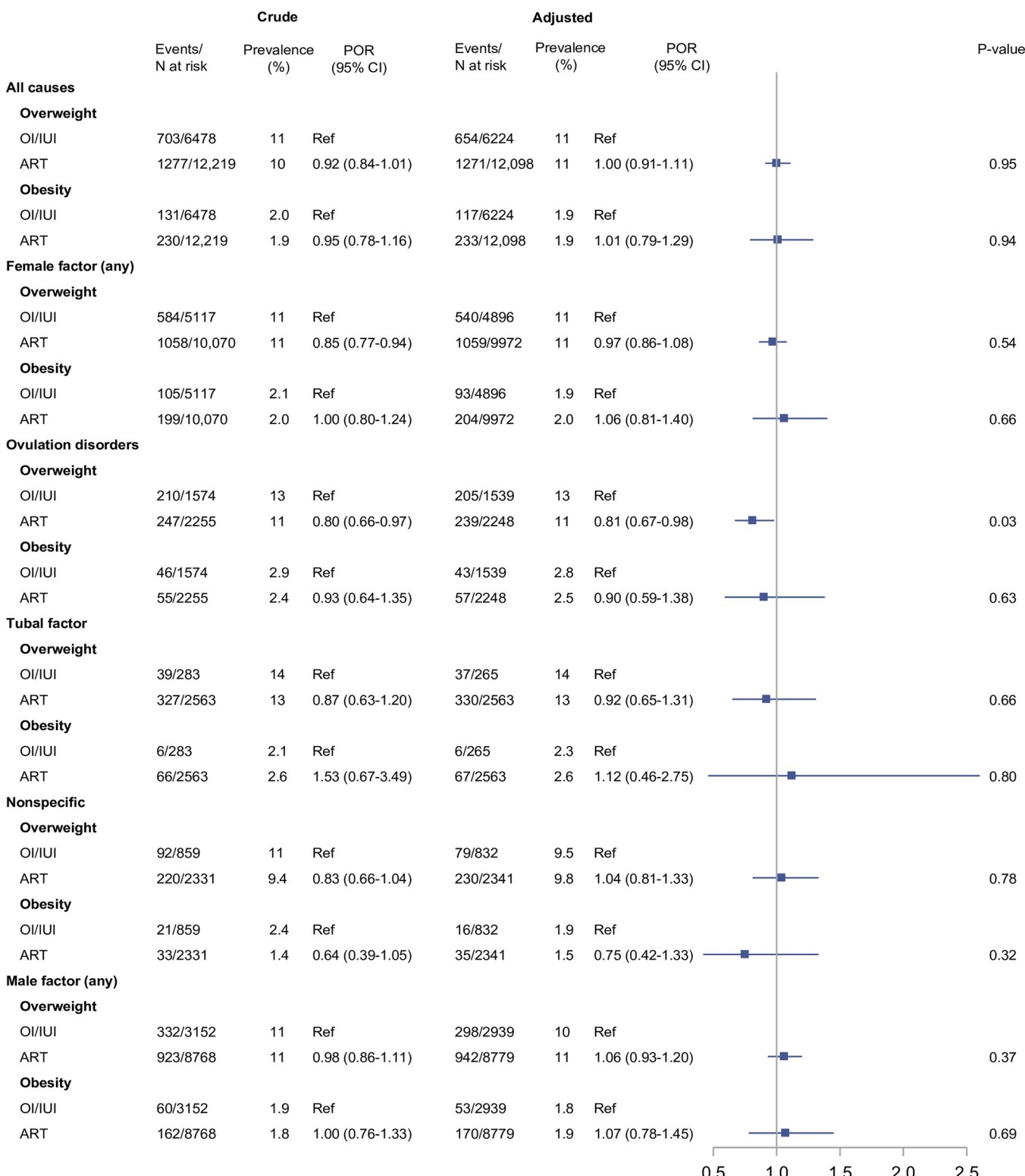

**Fig 2. Overweight and obesity in children born after ARTs compared to OI with or without IUI (OI/IUI), overall and within subgroups defined by underlying causes of infertility. Notes:** We adjusted for parental causes of infertility, maternal and paternal age at conception, maternal and paternal highest educational level at conception, maternal country of origin, maternal BMI, maternal smoking status, maternal and paternal hyperlipidemia/use of lipid-modifying drugs, maternal and paternal hypertension/use of antihypertensive drugs, diabetes (type I or II) diagnosed at any time before conception, parity, and year of conception. *P* values were calculated by the large-sample Wald (*Z*) test. Abbreviations: ART, assisted reproductive technology; BMI, body mass index; CI, confidence interval; IUI, intrauterine insemination; OI, ovulation induction; POR, prevalence odds ratio.

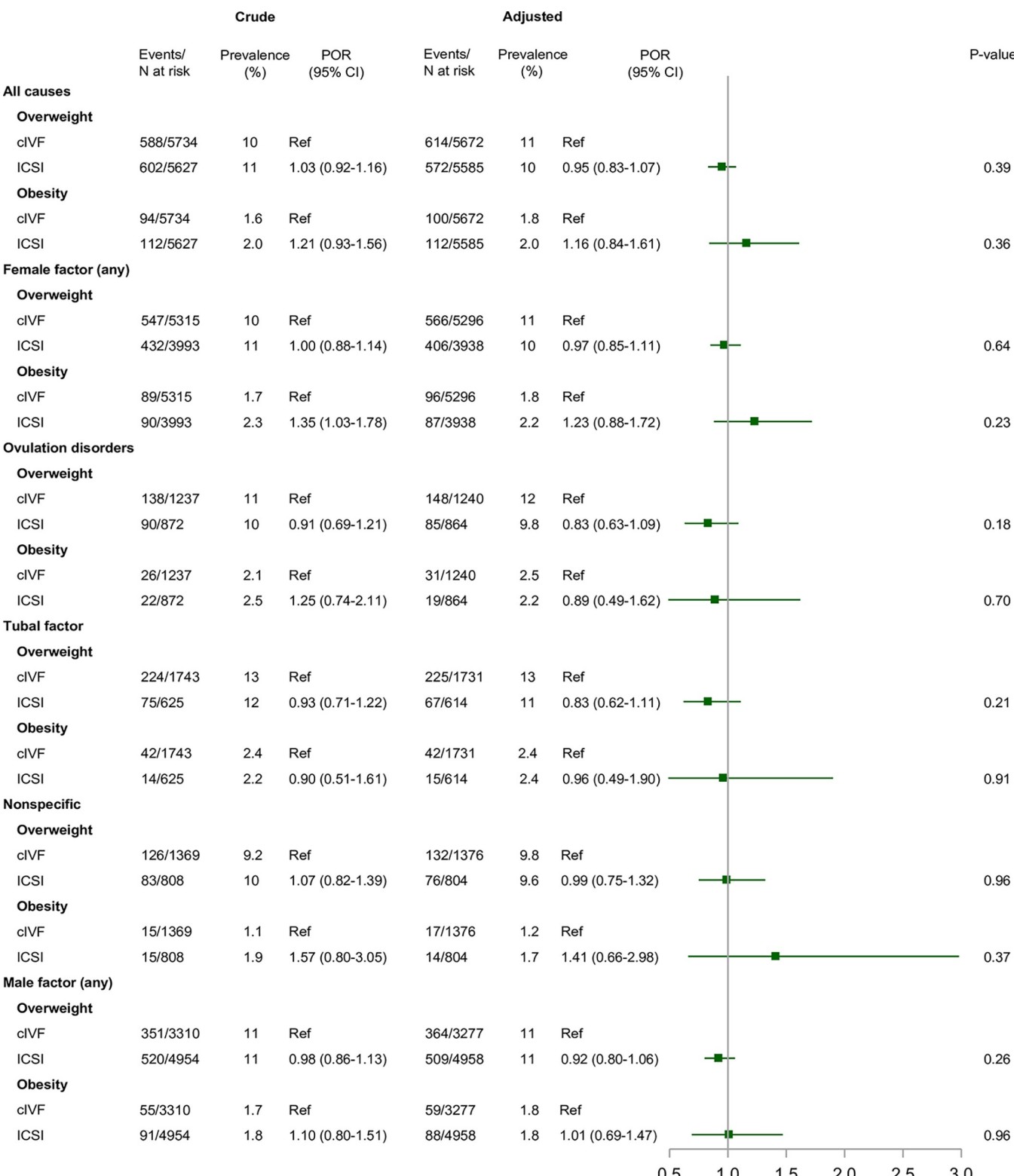

**Fig 3. Overweight and obesity in children born after ICSI compared to cIVF, overall and within subgroups defined by underlying causes of infertility.**
**Notes:** We adjusted for parental causes of infertility, maternal and paternal age at conception, maternal and paternal highest educational level at conception, maternal country of origin, maternal BMI, maternal smoking status, maternal and paternal hyperlipidemia/use of lipid-modifying drugs, maternal and paternal hypertension/use of antihypertensive drugs, diabetes (type I or II) diagnosed at any time before conception, parity, and year of conception. *P* values were calculated by the large-sample Wald (Z) test. Abbreviations: BMI, body mass index; CI, confidence interval; cIVF, conventional in vitro fertilization; ICSI, intracytoplasmic sperm injection; POR, prevalence odds ratio.

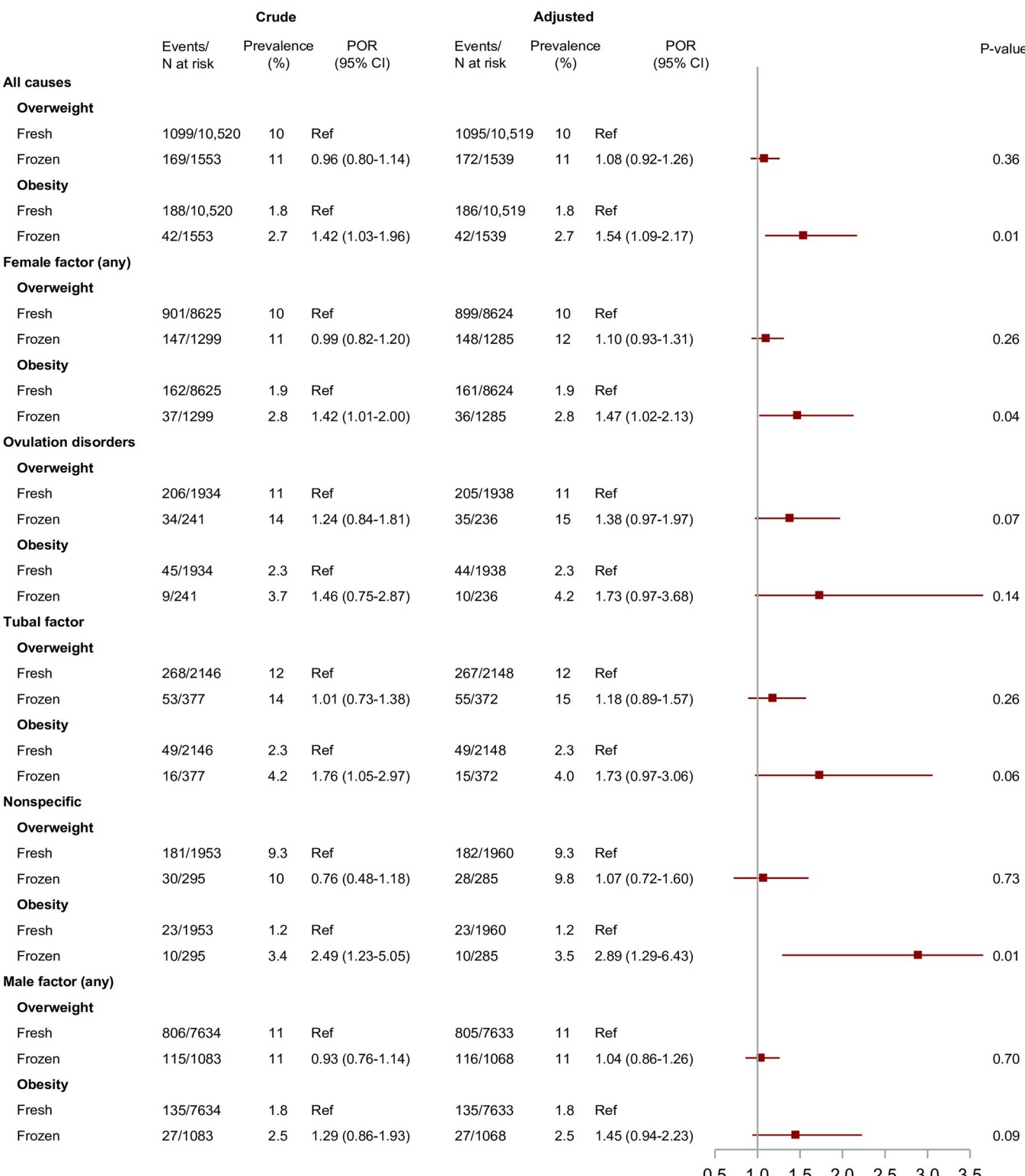

**Fig 4. Overweight and obesity in children born after frozen-thawed embryo transfer compared to fresh embryo transfer, overall and within subgroups defined by underlying causes of infertility. Notes:** We adjusted for parental causes of infertility, maternal and paternal age at conception, maternal and paternal highest educational level at conception, maternal country of origin, maternal BMI, maternal smoking status, maternal and paternal hyperlipidemia/use of lipid-modifying drugs, maternal and paternal hypertension/use of antihypertensive drugs, diabetes (type I or II) diagnosed at any time before conception, parity, and year of conception. Distribution of fresh vs. frozen embryo transfer by previous number of ART cycles: 0 cycles: 97.2% vs. 2.75%, 1 cycle: 78.9% vs. 21.1%, 2 cycles: 83.8% vs. 16.2%, ≥3 cycles: 84.5% vs. 15.5%. Distribution of fresh vs. frozen embryo transfer by maternal age (years): <25: 88.6% vs. 11.4%, 25–

29: 87.9% vs. 12.1%, 30–34: 87.2% vs. 12.7%, 35–39: 87.1% vs. 12.9%, ≥40: 88.3% vs. 11.7%. *P* values were calculated by the large-sample Wald (Z) test.
Abbreviations: ART, assisted reproductive technology; BMI, body mass index; CI, confidence interval; POR, prevalence odds ratio.

0.78) and frozen-thawed embryo transfer with OI/IUI (adjusted POR of obesity of 1.44, 95% CI: 0.96 to 2.17; *p* = 0.08) (S2, S3, and S4 Figs), respectively.

Sex of the child had a slight impact on the estimates, with a tendency towards higher point estimates in boys than in girls (S5 Text). Sensitivity analyses produced minor fluctuations in

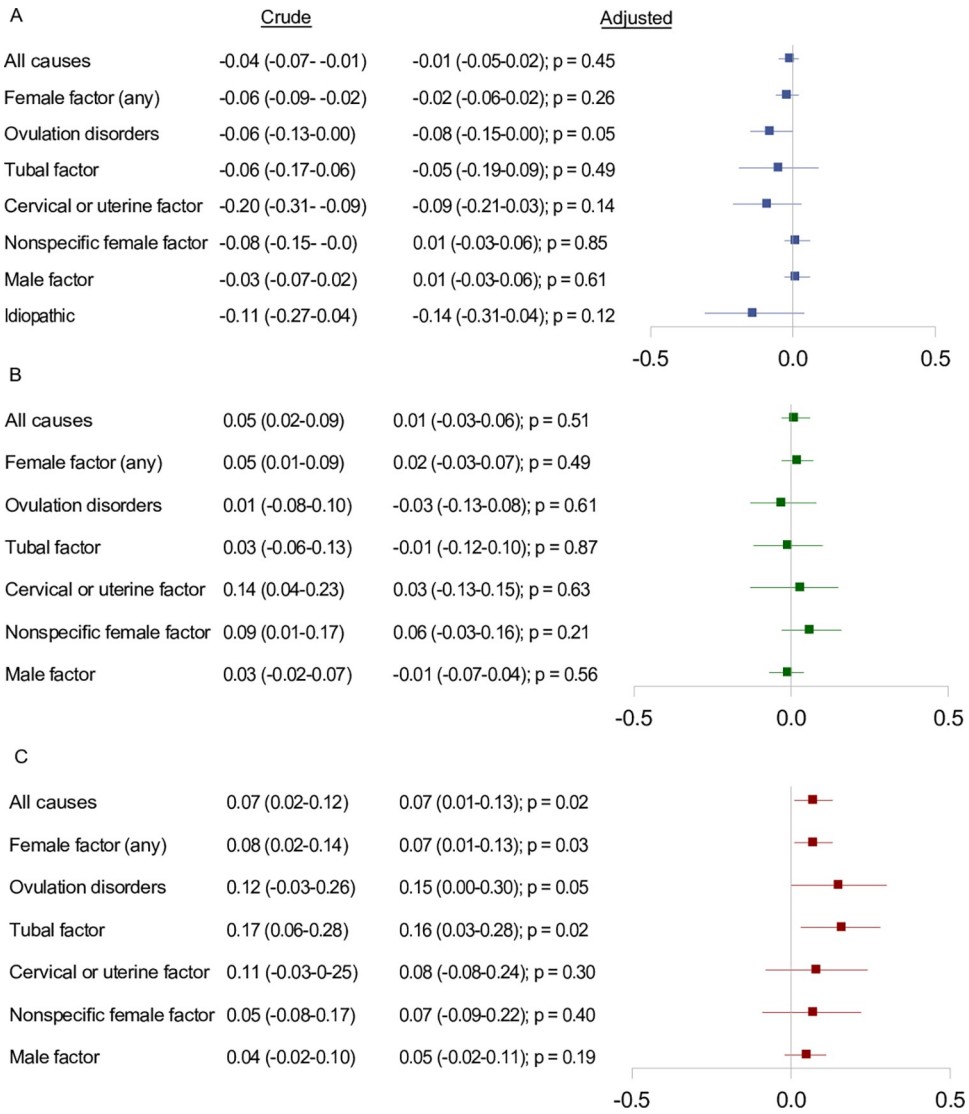

**Fig 5. Crude and adjusted differences in mean BMI Z-scores, overall and within subgroups defined by underlying causes of infertility. For children born after: (A) ARTs compared to OI with or without IUI (OI/IUI), (B) ICSI compared to cIVF, and (C) frozen-thawed embryo transfer compared to fresh embryo transfer. Notes:** We adjusted for parental causes of infertility, maternal and paternal age at conception, maternal and paternal highest educational level at conception, maternal country of origin, maternal BMI, maternal smoking status, maternal and paternal hyperlipidemia/use of lipid-modifying drugs, maternal and paternal hypertension/use of antihypertensive drugs, diabetes (type I or II) diagnosed at any time before conception, parity, and year of conception. *P* values were calculated using the large-sample Wald (Z) test. Abbreviation: ART, assisted reproductive technology; BMI, body mass index; CI, confidence interval; cIVF, conventional in vitro fertilization; IUI, intrauterine insemination; OI, ovulation induction.

estimates without consequences for our overall interpretation (S6 Text). Frozen-thawed embryo transfer in modified natural cycle (using hCG trigger only) and programmed cycle (estrogen and progesterone) yielded similar results (S6 Text). The supplemental analyses, using the general population comparator, provided similar direction of estimates as the main analyses (S7 and S8 Texts).

## Discussion

This nationwide population-based cohort study found no association with BMI at age 5 to 8 years when comparing ART versus OI/IUI and ICSI versus conventional IVF. However, use of frozen-thawed embryo transfer was associated with an excess risk of obesity compared to fresh embryo transfer and OI/IUI, respectively. Most associations appeared to be only modestly influenced by the underlying cause of infertility (no evidence of substantial effect modification).

This study expands prior knowledge in multiple ways. Importantly, the reasons that couples seek or receive fertility treatments could impact child outcomes, i.e., the indication for treatment itself could be influential. To address this concern, we only compared children born to parents receiving some form of fertility treatment. Hence, we reduced confounding from parental health, infertility, and ability/willingness to access fertility treatment. Moreover, the large study size, population approach, and use of medical registries with detailed clinical information allowed us to perform analyses by specific ART procedures and underlying causes of infertility. To provide information comparable to those of some prior studies, we also described outcomes among children born after fertility treatment and among children born after no fertility treatment, finding the first group at lower risk of obesity at age 5 to 8 years. This is consistent with results from 2 recent large cohort studies conducted by Elhakeem and colleagues (N = 158,066, of which 4,329 were born after ART) and Magnus and colleagues (N = 81,461, of which 1,721 were born after ART) [20,21].

We found a consistent positive association between frozen-thawed embryo transfer and obesity. Similarly, Magnus and colleagues reported higher mean BMI Z-scores for children born after frozen-thawed versus fresh embryo transfer up to age 6, but no difference at age 17 [21]. Hann and colleagues reported an adjusted mean BMI difference of 0.043 kg/m$^2$ (95% CI: −0.150 to 0.233) at age 5 year for children born after frozen-thawed (n = 1,091) versus fresh embryo transfer (n = 4,127) [6]. While frozen-thawed embryo transfer was associated with an elevated risk of obesity compared with fresh embryo transfer, we found that the crude risk was close to that observed in children from the general population born after no fertility treatment. This result was supported by Elhakeem and colleagues and Magnus and colleagues [20,21]. Former studies reported different obstetric and perinatal outcomes depending on the specific IVF protocols. As example, Busnelli and colleagues and Versen-Höynck and colleagues found higher risk of hypertensive disorders of pregnancy, preeclampsia, postpartum haemorrhage, and cesarean section when comparing frozen-thawed embryo transfer in programmed versus natural cycle [47,48]. In our study, we found similar associations for both types of endometrial preparation.

For ICSI, we confirmed the null association observed in prior studies [3,6,10,11,13,16,21]. Our observation of a tendency towards higher point estimates in boys than in girls is somewhat unsupported in the literature, but existing evidence is sparse [14,20,21]. Still, sex-specific differences in metabolic programming is widely acknowledged [46,49].

Our study has several potential limitations. First, approximately 13% of eligible children lacked outcome measurements. Recording of weight and height in the Danish National Children's Database comes from routine examinations of all Danish children, but missing

information occurs due to technical issues, such as randomly occurring local breakdowns of reporting systems, or removal of implausible measurements as part of a central validation process. Importantly, we found that the occurrence of missing BMI measurement was similar across all exposure and comparison groups. In addition, our evaluation of the data indicated that censoring would be noninformative. Second, our findings should be interpreted in light of restriction to live born children. If our exposures affected the probability of live birth, this restriction could lead to live birth bias in the presence of unmeasured common causes of fetal loss and high BMI [50]. It is possible that the probability of having a live birth differed across the exposure and comparison cohorts. Because of lack of data not resulting in livebirths, we were unable to examine this issue. Third, with respect to confounding, we used a comparison cohort of children born to parents receiving fertility treatment. Characteristics of parents in the ART and OI/IUI cohorts were homogenous with respect to measured confounders and plausibly also were somewhat alike with respect to unmeasured or unknown confounders. Still, we do not have data on the unmeasured or unknown, thus cannot rule out confounding. One example of a potentially important poorly measured factor is the severity of infertility. Fourth, we cannot dismiss nondifferential misclassification of treatments, outcomes, or covariates. Danish children born following fertility treatment abroad (i.e., reproductive tourism) would be misclassified as being born after no fertility treatment. Further, children born after no fertility treatment could have parents with infertility. The Danish registries do not capture time to pregnancy, but 5.4% of children in this group had parents with a prior diagnosis of infertility. Some children in the cohort were born after fertility treatment using donor oocytes or sperm. For these, the listed baseline characteristics concerns the registered parents, not the biological donor. While this recording should be appropriate for some characteristics (i.e., parental age, educational level, and smoking), it might be less appropriate for other traits or diagnoses with a genetic component. Fifth, we lacked detailed technical information about specific procedures, such as those used for freezing and thawing. Finally, our study did not explore the risk of obesity in adolescence or adulthood, or the development of metabolic diseases. Body fat composition and other elements of the metabolic syndrome are essential to consider [20,21].

Our overall null results provide reassuring results for couples with infertility seeking help. Our finding of elevated obesity risk after frozen-thawed versus fresh embryo transfer could reflect multiple pathways. First, gametes and embryos are exposed to hormones, in vitro manipulation, and culturing in artificial media during ART [7]. Such stimuli may lead to epigenetic programming of the metabolism of children, with a net result of increased obesity risk in a later obesogenic environment [51,52]. The potential impact of epigenetic programming may differ in frozen-thawed and fresh embryos due to differences in the in vitro manipulation techniques, and hormonal and intrauterine environments [51]. Also, the process of freezing and thawing might induce molecular changes in the embryo. Second, accumulating evidence consistently links the use of frozen-thawed embryos to larger weights at birth, which may persist into later life. Intrauterine growth conditions could be affected by various mechanisms, such as placental function, growth signalling pathways, or pregnancy complications (i.e., gestational diabetes). Third, couples with certain unknown or unmeasured traits or characteristics could be more likely to progress from fresh to frozen-thawed embryo treatment. Hence, confounding from such background factors is another possible explanation to the findings. Finally, given the multiple comparisons, our result may be a chance finding and thus needs confirmation in future studies.

In conclusion, our study found no evidence supporting an association between ART and BMI at age 5 to 8 years in a population of children born to parents treated for infertility. However, longer follow-up and assesment of other adverse metabolic outcomes need consideration.

Use of frozen-thawed embryo transfer was associated with an excess risk of obesity compared to fresh embryo transfer and OI/IUI, respectively. Despite elevated relative risks, the absolute risk differences were low. Reasons that underly this finding requires more investigation.

## Supporting information

**S1 STROBE Checklist. STROBE Statement—Checklist of items that should be included in reports of** *cohort studies.*
(PDF)

**S1 Text. Original analysis plan.**
(PDF)

**S2 Text. Characteristics of children with and without routine anthropometric evaluation at age 5–8 years.**
(PDF)

**S3 Text. Definition of exposure and covariates.**
(PDF)

**S4 Text. Crude and adjusted prevalence odds ratios (PORs) between covariates and overweight and obesity.**
(PDF)

**S5 Text. Overweight and obesity in children born after assisted reproductive technologies, stratified by sex of the child.**
(PDF)

**S6 Text. Results from sensitivity analyses.**
(PDF)

**S7 Text. Crude and adjusted prevalences (%) and prevalence odds ratios (PORs) for overweight and obesity in children born after assisted reproductive technologies (ARTs) vs. children from the general population.**
(PDF)

**S8 Text. Crude and adjusted differences in mean BMI Z-scores comparing children born after assisted reproductive technologies (ARTs) vs. children from the general population.**
(PDF)

**S1 Fig. Propensity score distributions across exposure and comparison cohorts overall and for subgroups defined by underlying cause of infertility for (A) ARTs vs. OI with or without IUI (OI/IUI), (B) ICSI vs. cIVF, and (C) frozen-thawed vs. fresh embryo transfer. Notes:** Abbreviations: ART, assisted reproductive technology; cIVF, conventional in vitro fertilization; ICSI, intracytoplasmic sperm injection; IUI, intrauterine insemination; OI; ovulation induction.
(PDF)

**S2 Fig. Overweight and obesity in children born after fresh embryo transfer compared to OI with or without IUI (OI/IUI), overall and within subgroups defined by underlying causes of infertility. Notes:** We adjusted for parental causes of infertility, maternal and paternal age at conception, maternal and paternal highest educational level at conception, maternal country of origin, maternal BMI, maternal smoking status, maternal and paternal hyperlipidemia/use of lipid-modifying drugs, maternal and paternal hypertension/use of antihypertensive drugs, diabetes (type I or II) diagnosed at any time before conception, parity, and year of

conception. *P* values were calculated by the large-sample Wald (Z) test. Abbreviations: BMI, body mass index; CI, confidence interval; IUI, intrauterine insemination; OI, ovulation induction; POR, prevalence odds ratio.
(PDF)

**S3 Fig. Overweight and obesity in children born after frozen-thawed embryo transfer compared to OI with or without IUI (OI/IUI), overall and within subgroups defined by underlying causes of infertility. Notes:** We adjusted for parental causes of infertility, maternal and paternal age at conception, maternal and paternal highest educational level at conception, maternal country of origin, maternal BMI, maternal smoking status, maternal and paternal hyperlipidemia/use of lipid-modifying drugs, maternal and paternal hypertension/use of anti-hypertensive drugs, diabetes (type I or II) diagnosed at any time before conception, parity, and year of conception. All standardized differences were ≤0.15 after weighting, except for maternal BMI, which yielded an standardized difference of 0.29. *P* values were calculated using the large-sample Wald (Z) test. Abbreviations: BMI, body mass index; CI, confidence interval; IUI, intrauterine insemination; OI, ovulation induction; POR, prevalence odds ratio.
(PDF)

**S4 Fig. Crude and adjusted differences in mean BMI Z-scores, overall and within subgroups defined by underlying causes of infertility. For children born after: (A) fresh embryo transfer compared to OI with or without IUI (OI/IUI) and (B) frozen-thawed embryo transfer compared to OI/IUI. Notes:** We adjusted for parental causes of infertility, maternal and paternal age at conception, maternal and paternal highest educational level at conception, maternal country of origin, maternal BMI, maternal smoking status, maternal and paternal hyperlipidemia/use of lipid-modifying drugs, maternal and paternal hypertension/use of antihypertensive drugs, diabetes (type I or II) diagnosed at any time before conception, parity, and year of conception. All standardized differences were ≤0.15 after weighting, except for maternal BMI, which yielded an standardized difference of 0.29. *P* values were calculated by the large-sample Wald (Z) test. Abbreviation: BMI, body mass index; CI, confidence interval; IUI, intrauterine insemination; OI, ovulation induction.
(PDF)

## Author Contributions

**Conceptualization:** Kristina Laugesen, Katalin Veres, Sonia Hernandez-Diaz, Yu-Han Chiu, Anna Sara Oberg, John Hsu, Paolo Rinaudo, Mandy Spaan, Flora van Leeuwen, Henrik Toft Sørensen.

**Data curation:** Kristina Laugesen, Katalin Veres, Henrik Toft Sørensen.

**Formal analysis:** Katalin Veres.

**Funding acquisition:** Sonia Hernandez-Diaz, Yu-Han Chiu, John Hsu.

**Investigation:** Kristina Laugesen, Sonia Hernandez-Diaz, Yu-Han Chiu, Anna Sara Oberg, John Hsu, Paolo Rinaudo, Mandy Spaan, Flora van Leeuwen, Henrik Toft Sørensen.

**Methodology:** Kristina Laugesen, Katalin Veres, Sonia Hernandez-Diaz, Yu-Han Chiu, Anna Sara Oberg, John Hsu, Paolo Rinaudo, Mandy Spaan, Flora van Leeuwen, Henrik Toft Sørensen.

**Project administration:** Kristina Laugesen, Henrik Toft Sørensen.

**Supervision:** Katalin Veres, Sonia Hernandez-Diaz, Yu-Han Chiu, Anna Sara Oberg, John Hsu, Paolo Rinaudo, Mandy Spaan, Henrik Toft Sørensen.

**Validation:** Katalin Veres, Sonia Hernandez-Diaz, Yu-Han Chiu, Anna Sara Oberg, John Hsu, Paolo Rinaudo, Mandy Spaan, Henrik Toft Sørensen.

**Visualization:** Kristina Laugesen, Katalin Veres.

**Writing – original draft:** Kristina Laugesen, Sonia Hernandez-Diaz, Yu-Han Chiu, Anna Sara Oberg, John Hsu, Paolo Rinaudo, Mandy Spaan, Henrik Toft Sørensen.

**Writing – review & editing:** Kristina Laugesen, Katalin Veres, Sonia Hernandez-Diaz, Yu-Han Chiu, Anna Sara Oberg, John Hsu, Paolo Rinaudo, Mandy Spaan, Flora van Leeuwen, Henrik Toft Sørensen.

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
