## [Editor Report · Decision Letter 0]

25 Apr 2023

Dear Dr Laugesen, 

Thank you for submitting your manuscript entitled "Overweight or obesity in children born after assisted reproductive technologies: A population-based cohort study" for consideration by PLOS Medicine.

Your manuscript has now been evaluated by the PLOS Medicine editorial staff and I am writing to let you know that we would like to send your submission out for external peer review.

Please re-submit your manuscript within two working days, i.e. by Apr 27 2023 11:59PM.

Kind regards,

Philippa Dodd, MBBS MRCP PhD

PLOS Medicine

---

## [Decision Letter · Decision Letter 1]

30 Jun 2023

Dear Dr. Laugesen,

Thank you very much for submitting your manuscript "Overweight or obesity in children born after assisted reproductive technologies: A population-based cohort study" (PMEDICINE-D-23-01124R1) for consideration at PLOS Medicine. 

Your paper was discussed with an academic editor with relevant expertise and sent to independent reviewers, including a statistical reviewer. The reviews are appended at the bottom of this email and any accompanying reviewer attachments can be seen via the link below:

[LINK]

In light of these reviews we will not be able to accept the manuscript for publication in the journal in its current form, but we would like to invite you to submit a revised version that addresses the reviewers' and editors' comments fully. You will appreciate that we cannot make a decision about publication until we have seen the revised manuscript and your response, and we expect to seek re-review by one or more of the reviewers. 

We hope to receive your revised manuscript by Jul 24 2023 11:59PM. Please email us (plosmedicine@plos.org) if you have any questions or concerns.

Please let me know if you have any questions, and we look forward to receiving your revised manuscript. 

Sincerely,

Richard Turner PhD, for Louise Gaynor-Brook, MBBS PhD

Consulting editor, PLOS Medicine

plosmedicine@plos.org

Please add "in Denmark" or similar to the title. 

In the abstract, please add a new final sentence to the 'Methods and findings' subsection, which should begin "Study limitations include ..." or similar and should quote 2-3 of the study's main limitations. 

Where available, please quote p values alongside 95% CI. 

After the abstract, please add a new and accessible three-subsection 'Author summary' section in non-identical prose. You may find it helpful to consult one or two recently published research papers in PLOS Medicine to get a sense of the preferred style. 

Please state in the Methods section (Main text) whether or not the study had a protocol or prespecified analysis plan, and if so attach the document(s) as a supplementary file, referred to in the text. 

As you have done an observational study, we ask you to avoid the word "effect" when describing your findings. For example, at line 63 that could become "no impact" or "no association between". 

Please ensure that the form "age 5-8 years" is used consistently throughout; likewise, "follow-up" where appropriate.

While numbers should be spelt out at the start of sentences, please use numerals throughout the text (e.g., "... from 6 countries" at line 197. 

Please move the information on ethics approval from the end of the main text to the Methods section. 

Please remove the information on competing interests, funding and data sharing from the end of the main text. In the event of publication, this information will appear in the article metadata, via entries in the submission form. 

Throughout the text, please adapt reference call-outs to journal style: at line 85, for example, as "... past 4 decades [1,2]." (noting the absence of spaces within the square brackets). 

In the reference list, please use the journal name abbreviations "PLoS Med.", "PLoS ONE" and "BMJ".

Noting reference 1, we suggest spelling out the group author name. 

Please add full access details to reference 25.

Please include a completed checklist for the most appropriate reporting guideline as a supplementary file, labelled "S1_RECORD_Checklist" or similar and referred to as such in the Methods section (main text). 

In the checklist, please refer to individual items by section (e.g., "Methods") and paragraph number, not by page or line numbers as these generally change in the event of publication.

Comments from the reviewers:

*** Reviewer #1: 

This manuscript provides data on association of ART and adiposity in a Danish cohort

- The study cohort is unique and provides a valuable opportunity for these analyses

- Analyses are well designed and performed

- Results are important especially for public health purposes

*** Reviewer #2: 

This paper investigates, in a population based registry study, if ART is associated with overweight and obesity in children born after ART in comparison to children born after non in-vitro fertilization fertility treatment (NIFT) and children born after spontaneous conception. They found that children born after ART in general had similar risk of overweight and obesity as children born after NIFT while children born after frozen embryo had a higher risk for obesity.

This is an important topic, in view of the increasing number of children born after ART worldwide and particularly after frozen embryo transfer.' The ms is well written and Conclusion is careful.

I have the following specific comments:

1.Children were between 5-8 years of age. How can it be that large range? Children in Denmark start (compulsory) pre-school the year they will be 6. Isn't there a more exact age when children go through these examinations? Examinations are performed at "primary school" we are told. Does that include the preschool year? Further how is the different age of the children managed in the main analyses? Even though median age is similar.

2. How about children who don't attend primary school? I presume that the 13% missing is among those attending primary school or? Reasons for children who don't attend primary school might be neurological/neurodevelopment diseases-more prevalent in ART than spontaneous conception. How large part of children do not attend primary school in the three groups? And reasons for that? Thus, can there be a selection of more "healthy" ART children in the ART group? Further, you mention that missing examination is at random. How do you know that? Might it be that obese children refuse these examination more often than normal weight children?

Thus, there are two ways of missing, those who don't attend primary school and those refusing examination. You should separate these and give us more explanations.

3. I question to say that you investigate "the effect of ART"…. (line 179(. This is an observational study where causality can not be established. The word "association " fits better.

4. I don't think you compare ART with non IVF fertility treatment. NIFT includes only ovulation induction and intrauterine inseminations. There are other kind of fertility treatment and thus not included ie surgery for myomas, uterine septas, weight loss for ovulation disorders etc. There must also be patients registered with a diagnosis of infertility but conceived spontaneously. I presume these latter are in the SC group. Should be discussed and as a limitation. Better to say that you compare ART to children born after OI and IUI.

5. How are patients, receiving ART treatment and NIFT abroad ("reproductive tourism") and delivering in Denmark managed? How big part do they constitute? And people coming from abroad, receiving fertility treatment in Denmark but not delivering there. I understand they must be excluded but this should be clarified. Reproductive tourism is quite substantial in Denmark.

6. Nothing is written about oocyte donation and sperm donation cycles? Are these included or excluded? Should, if excluded be presented in flow chart. If included- how do you know parental characteristics of these donors? This part needs to be clarified.

7. How were covariates identified? By statistical analyses? From the literature?-please in that case give references. 

8.Multiplicity is not mentioned? I presume this cohort includes singletons and multiples-these are more frequent in ART than SC. Multiple birth children have many different characteristics compared to singletons-known and unknown. Adjustment for multiples? Or a sensitivity analysis for singletons and /or stratification? Even though the propensity score analysis partly (probably?) takes care of that.

9. Maybe avoid words like "similar", higher etc when no statistical comparison have been made. Since I presume you regard these populations as samples. (if not no statistical comparison at all is needed). Several places in the ms.

10. Table 2. Aren't there any missing for paternal BMI? I presume it is "child from a singleton pregnancy" not singleton pregnancies

11. In the paragraph "Overweight and obesity among children aged 5-8 born after fertility treatment and no fertility treatment" (there are no lines inserted any longer!) it is unclear when crude or adjusted results are presented. Can you clarify! In which table is this shown? Table number should be referred in text.

12. suppl Table. Little strange to give Europe as ref and not Nordic? Reason? And does European mean Europe except Nordic countries?

13. Discussion

It must be wrong , in start of Discussion, sec paragraph, writing that children from no infertility treatment have a lower risk of obesity than ART. It is the other way around.

14. Can you give a possible mechanism for the higher prevalence of obesity in children from frozen embryo transfer. Except speaking about "epigenetics" which myself I am quite tired of.

15. I don't understand you first limitation "First, its setting was within a universal tax supported healthcare system with free access to fertility treatment". But ALL IVF children are stated to be included, from publicly funded as well as private units. What is meant?

*** Reviewer #3: 

Thanks for the opportunity to review your manuscript. My role is as a statistical reviewer, so my review concentrates on the study design, data, and analysis that are presented. I have put general questions first, followed by queries relevant to a specific section of the manuscript (with a line reference).

This manuscripts presents a large cohort study, comparing BMI (as either overweight, or obese) between children (5-8 years) born after assisted reproductive technologies (IVF and similar methods) with non-IVF technologies (ovulation induction etc.), and made comparisons between specific ART and NIFT procedures. The rationale for comparing the two categories of technology is to have a similar set of parents who have fertility issues, removing a potential source of confounding if children born after ART are compared with those born to parents who did not receive either ART or NIFT. Data comes from several linked registries that capture information on births, fertility treatment, height and weight of children, as well as inpatient visits, dispensing of medication, and demographic and civil registries. The data for the main analyses included ~327,000 children, after excluding children who did not have height and weight recorded (13% of all livebirths identified, and consistent across categories of reproductive treatment). Out of all the characteristics examined, only country of origin (with more missing/unknown for this variable in those without height/weight measurements). Data from school measurements was taken to derive BMI, which as then classified as overweight/obese based on IOTF centiles, and additionally as a continuous variable converted to a z-score (WHO). The main comparisons is children born after ART vs. NIFT, and there are secondary analyses comparing among types of ART, and fresh vs. frozen embryo transfers. A propensity score for type of reproductive therapy was estimated with logistic regression, using a wide range of relevant covariates (with information about their derivation detailed in the supplementary appendix), but not post-birth variables. The PS was then used to create IPTW weights in the substantive analysis. Standardised differences between groups in the weighted dataset were small. A range of sensitivity analyses were considered and the rationale for these was explained. The limitations section in the discussion is appropriate for the study design and data. 

I enjoyed reading this manuscript - the methodology is clear and appropriate and so is the rationale. I was amazed that the data used here is routinely collected - and a bit jealous.

After estimating the propensity score, was there evidence of common support? A figure for each analysis in the appendix that shows the distribution in scores across the relevant groups would be most effective way to demonstrate this.

L61. This sentence was unclear - it might be better to say directly what the 'associations' is, e.g. association between different fertility treatments and overweight/obesity. 

L252. Why was GEE used here? Was this to account for correlation between children born to the same parents? 

L253. I wasn't clear here why a prevalence odds ratio was estimated. I have typically seen this term used when the study design means it was not possible to distinguish between incident and prevalent cases when the outcome is measured. With your study design, effectively all overweight/obesity is incident as the outcome is based on the first measurement recorded at school. 

*** Reviewer #4: 

General comments

This is a nationwide cohort study on BMI after different ART treatments including a cohort more than 300.000 children, whereof 113.675 were born after ART and 7728 after non-invasive fertility treatment (NIF). The authors compared the effect of different fertility treatments on BMI in children at ages 5-8 years, adjusting for and stratifying by causes of parental infertility. Crude and adjusted (using stabilized inverse probability of treatment weights) prevalences were calculated with overweight or obesity at ages 5-8 years, prevalence odds ratios (PORs) and differences in mean BMI z-scores. This paper deals with an intriguing topic and is a well-conducted and well- written study however there are some concerns that need to be addressed and still needs some revision. The study comprises a large cohort of more than 300.000 children born from 2007 to 2012 including ART and children born after non-invasive fertility treatment (NIFT) and the paper adds to the accumulating knowledge on BMI after various ART procedures. 

* In the introduction a short section on the risk of small-for-gestational age (SGA) and large-for-gestational-age (LGA) in children from fresh and frozen embryo transfer respectively should be added. 

* The authors conclude that there was no effect of ART vs. NIFT on BMI at ages 5-8 years. BMI was comparable for ICSI vs. conventional IVF. However, use of frozen-thawed embryo transfer was associated with a 1.5-fold increased risk of obesity at ages 5-8 compared to fresh embryo transfer.

The authors should make a specific analysis comparing FET vs. natural conception, as FET children are born with higher mean birth weight and are more likely to be LGA, the opposite is the case with children from fresh embryo transfer that are born with a lower mean BW and are more likely to be SGA. The BW after fresh and FET pointing in opposite directions may be translated into BMI at 5-8 years of age. Therefore, FET children should be compared with a more "neutral" control group than the children after fresh embryo transfer and this should be the group with no fertility treatment, which is also written in page 20 but this should be emphasized more instead of the comparison with the fresh embryo transfer group. The conclusion in abstract and in the main paper should be modified accordingly. 

* Regarding Table I, it is not obvious that out of 2,485 with ovulatory 58.9% were treated with ART. This needs to be specified.

* A comparison of LGA in the FET vs. the general population should be performed. 

* For all sub-analyses with FET, these should also be performed comparing with children from the general population. 

* Discussion: It is not correct to highlight that the use of frozen-thawed embryo transfer was associated with a 1.5-fold increased risk of obesity at ages 5-8 compared to fresh embryo transfer without mentioning the comparisons to the general population. 

* In the discussion a section on the different treatment modalities in fresh vs. frozen should be included, further a discussion on the programmed cycle FET with sequential estradiol and progesterone (and no corpus luteum) should be mentioned showing higher rates of preeclampsia and LGA (Busnelli et al., Hum Reprod 2022 and von versen-hoynck, Hypertension 2019). 

* If the authors were able to access data on medication in the IVF registry and was able to pin-point the various endometrial preparation protocols in FET cycles that would be very interesting. These data are available.

***

[LINK]

---

## [Decision Letter · Decision Letter 2]

25 Sep 2023

Dear Dr. Laugesen,

Thank you very much for submitting your manuscript "Overweight or obesity in children born after assisted reproductive technologies in Denmark: A population-based cohort study" (PMEDICINE-D-23-01124R2) for consideration at PLOS Medicine. 

Your paper was re-evaluated by three reviewers, including a statistical reviewer, and discussed among all the editors here. The reviews are appended at the bottom of this email and any accompanying reviewer attachments can be seen via the link below:

[LINK]

In light of these reviews, I am afraid that we will not be able to accept the manuscript for publication in the journal in its current form, but we would like to consider a revised version that fully addresses the reviewers' and editors' comments. Obviously we cannot make any decision about publication until we have seen the revised manuscript and your response, and we plan to seek re-review by one or more of the reviewers. 

We expect to receive your revised manuscript by Oct 02 2023 11:59PM. Please email me (lgaynor@plos.org) if you have any questions or concerns.

We look forward to receiving your revised manuscript. 

Sincerely,

Louise Gaynor-Brook, MBBS PhD

lgaynor@plos.org

plosmedicine.org

Thank you for submitting your revised manuscript. We mirror the comments from Reviewer 2 and request a statistical comparison between ART and spontaneous conception in your revised manuscript, which will then be re-reviewed. 

Comments from the reviewers:

Reviewer #2: The authors have answered most comments satisfactory and revised the manuscript accordingly.

I have a few additional comments:

1.How could restriction to live births be a limitation? Dead children can not contribute to the follow up at 5-8 years.

2.You state that 0.1% do not attend primary school in Denmark without given a reference. In Norrman et al, Human Reprod 2018 2.8% did not attend primary school . i doubt that these figures would be that different between Sweden and Denmark. Can you clarify?

3. I miss a statistical comparison between ART and spontaneous conception (non infertile group). Both for ART in general and for frozen embryo transfer. I understand the authors arguments for not doing this but I still think it would strenghten the paper of several reasons. One for comparison with other studies. It can be added as a supplement and you can keep the main analysis as it is. In fact , most studies, while pinpointing the limitation, use general population as controls.

4. Even though there is an increased rate of obesity in children from frozen embryo transfer, the absolute increase is rather modest. I think that should be added in Conclusion (in abstract etc)

Reviewer #3: Thanks for the revised manuscript and responses my original questions. The revised manuscript covers all my original queries. I enjoyed reading this and also learning more about the level integration of health data in Denmark.

The adjustment to the ascertainment of DM is a good idea, I have similarly used the same strategy of only using the initiation of metformin as the start of DM when used in conjunction with other information from a study participant that occurs later. 

I agree that putting a post-exposure variable in the analyses would not be good idea as it could create collider-stratification bias. 

Reviewer #4: The authors have responded thoroughly to all the reviewers comments and it is ready for publication.

[LINK]

---

## [Decision Letter · Decision Letter 3]

8 Nov 2023

Dear Dr. Laugesen,

Thank you very much for re-submitting your manuscript "Overweight or obesity in children born after assisted reproductive technologies in Denmark: A population-based cohort study" (PMEDICINE-D-23-01124R3) for review by PLOS Medicine.

I have discussed the paper with my colleagues and it was also seen again by two reviewers. I am pleased to say that provided the remaining editorial and production issues are dealt with we are planning to accept the paper for publication in the journal.

[LINK]

We expect to receive your revised manuscript within 1 week. Please email me (lgaynor@plos.org) if you have any questions or concerns.

We look forward to receiving the revised manuscript by Nov 15 2023 11:59PM.   

Sincerely,

Louise Gaynor-Brook, MBBS PhD

lgaynor@plos.org

plosmedicine.org

Requests from Editors:

Thank you for your patience with a longer assessment process than we anticipated, and apologies for the delay in providing you with an editorial decision. 

General comments:

Please avoid use of ‘effect’ throughout the main text

To help us extend the reach of your research, please provide any Twitter handle(s) that would be appropriate to tag, including your own, your coauthors’, your institution, funder, etc.

Data availability:

The Data Availability Statement (DAS) requires revision. Please provide contact information for data requests (web or email address). Please note that a study author cannot be the contact person for the data.

Abstract Methods and Findings:

Please provide brief demographic details of the study population (e.g. sex, etc)

Please include the important dependent variables that are adjusted for in the analyses.

Line 60 - please provide results for both overweight and obesity for ICSI vs IVF

Author Summary:

Line 85 - please revise to …‘the association between in vitro fertilization and Body Mass Index…’

Under the subheading ‘What did the researchers do and find?’, please add a bullet point explaining the results of the comparison between ART and the general population

In a final bullet point under the subheading ‘What Do These Findings Mean?’, please describe the main limitations of the study in non-technical language.

Introduction:

Line 137-152 - when discussing previous studies, please be clear that the evidence generated is observations, using terms such as associations, risk, etc. For example, “Recent studies have found an association between ART and lower BMI…” etc. 

Line 154 - please revise to “...compare BMI among children aged 5-8 years…” 

Methods:

Please refer to your prospective protocol/analysis plan early in the Methods section (S1 Text).Changes in the analysis (including those made in response to peer review comments) should be identified as such in the Methods section of the paper, with rationale. 

Line 224 - Please replace "subject" with participant, patient, individual, or person.

Line 227 - please avoid use of ‘effect’; we suggest “This comparison reflects the combination of the treatment… and not solely the treatment” or similar.

Lines 318-321 - please avoid use of ‘effect’; we suggest “...to enable comparison between…” 

Line 332 - we suggest revising to “..to estimate the average effect of treatment…”

Results: 

For the adjusted analyses in Fig. 5, please also provide the unadjusted analyses.

Figures (including in the supplementary files):

When a p value is given, please specify the statistical test used to determine it in the figure legend. 

Please define all abbreviations used in the figure legend of each figure.

Please provide the unadjusted comparisons in Fig S4.

Tables (including in the supplementary files):

When a p value is given, please specify the statistical test used to determine it.

Please define abbreviations used in the table legend of each table.

Please provide the unadjusted comparisons as well as the adjusted comparisons in S7 Text and S8 Text.

References:

Please ensure that journal name abbreviations match those found in the National Center for Biotechnology Information (NCBI) databases (http://www.ncbi.nlm.nih.gov/nlmcatalog/journals), and are appropriately formatted and capitalised e.g. Ref 1 should be Hum Reprod Open

Please also see https://journals.plos.org/plosmedicine/s/submission-guidelines#loc-references for further details on reference formatting. 

Comments from Reviewers:

Reviewer #2: I think the authors have answered all my comments and revised the paper accordingly. Thank you! I have no further comments.

Reviewer #3: The added analysis between ART and SC looks fine, and the explanation of the risk of the collider effect is good (that is a handy reference). All fine with the manuscript from my perspective.

[LINK]

---

## [Editor Report · Decision Letter 4]

16 Nov 2023

Dear Dr Laugesen, 

On behalf of my colleagues and the Academic Editor, Prof. Sarah Stock, I am pleased to inform you that we have agreed to publish your manuscript "Overweight or obesity in children born after assisted reproductive technologies in Denmark: A population-based cohort study" (PMEDICINE-D-23-01124R4) in PLOS Medicine.

Before your manuscript can be formally accepted you will need to complete some formatting changes and final editorial requests, which you will receive in a follow up email. Please be aware that it may take several days for you to receive this email; during this time no action is required by you. Once you have received these formatting requests, please note that your manuscript will not be scheduled for publication until you have made the required changes. All authors must complete the COI questionnaire prior to publication - please note that some responses are currently missing. 

PRESS

Sincerely, 

Louise Gaynor-Brook, MBBS PhD 

lgaynor@plos.org